# Genetic Analysis and Epitope Prediction of SARS-CoV-2 Genome in Bahia, Brazil: An In Silico Analysis of First and Second Wave Genomics Diversity

**Gabriela Andrade** [1], **Guilherme Matias** [1], **Lara Chrisóstomo** [1], **João da Costa-Neto** [1], **Juan Sampaio** [1], **Arthur Silva** [2] **and Isaac Cansanção** [1,*]

1 Collegiate of Medicine, Campus Paulo Afonso, Bahia, Federal University of the San Francisco Valley (UNIVASF), Paulo Afonso 58605780, Brazil

2 Collegiate of Natural Sciences, Campus Serra da Capivara, Piauí, Federal University of the San Francisco Valley (UNIVASF), São Raimundo Nonato 64770000, Brazil

* Correspondence: isaac.farias@univasf.edu.br

**Abstract:** COVID-19 is an infectious disease caused by SARS-CoV-2. This virus presents high levels of mutation and transmissibility, which contributed to the emergence of the pandemic. Our study aimed to analyze, in silico, the genomic diversity of SARS-CoV-2 strains in Bahia State by comparing patterns in variability of strains circulating in Brazil with the first isolated strain NC_045512 (reference sequence). Genomes were collected using GISAID, and subsequently aligned and compared using structural and functional genomic annotation. A total of 744 genomes were selected, and 20,773 mutations were found, most of which were of the SNP type. Most of the samples presented low mutational impact, and of the samples, the P.1 (360) lineage possessed the highest prevalence. The most prevalent epitopes were associated with the ORF1ab protein, and in addition to P.1, twenty-one other lineages were also detected during the study period, notably B.1.1.33 (78). The phylogenetic tree revealed that SARS-CoV-2 variants isolated from Bahia were clustered closely together. It is expected that the data collected will help provide a better epidemiological understanding of the COVID-19 pandemic (especially in Bahia), as well as helping to develop more effective vaccines that allow less immunogenic escape.

**Keywords:** COVID-19; coronavirus; genomics; variants; mutation; Bahia

## 1. Introduction

In December 2019, severe acute respiratory syndrome type 2 emerged as a pneumonia of unknown origin in Wuhan, Hubei Province, China. The pathogen was subsequently identified as SARS-CoV-2, and is responsible for causing the 2019 Coronavirus Disease (COVID-19). Due to its high transmissibility rate, the virus was soon disseminated in various other countries, which led the World Health Organization (WHO) to declare a state of public health emergency of international concern, and soon afterwards, a state of pandemic [1,2]. According to the WHO, as of 31 December 2022, the world has now presented more than 730 million confirmed COVID-19 infections and almost 6.7 million deaths (https://covid19.who.int/, accessed on 1 March 2023).

SARS-CoV-2 is the seventh human coronavirus (CoV) to be described. Its genome was first sequenced in January 2020 in China. This virus is a betacoronavirus from the Coronaviridae family, which usually cause respiratory and gastrointestinal tract problems [2,3]. Since the beginning of the COVID-19 pandemic, genetic analyses of SARS-CoV-2 in various countries and at different times have revealed that the virus has undergone many mutations that may well confer new chemical properties to its viral proteins. As examples, the D614G and 501Y mutations of the spike (S) protein confer greater transmissibility to the virus [4]. Various studies have identified SARS-CoV-2 genomic variations of differing

types, including missense, synonyms, insertions, deletions, and non-coding mutations [5]. This mutational variation has allowed the emergence of strains around the globe, such as B.1 (derived from the Chinese B variant), which resulted in a large Italian outbreak [6].

The pandemic has brought challenges that need to be addressed, and bioinformatics can play a fundamental role in this process. Obtaining and analyzing crucial parameters, including the virus transmissibility rate, reconstructing transmission routes, and identifying possible animal sources and reservoirs [7], are all necessary. In the specific case of this study, the objective was to conduct in silico analysis of the diversity of SARS-CoV-2 genomes circulating in the state of Bahia. Monitoring these factors is essential for many reasons and can help with genetic diversity analysis, associating clinical and epidemiological patterns, evaluating diagnostic methods, and developing more effective vaccine production or other therapeutic approaches [8]. Our specific emphasis was on the state of Bahia.

## 2. Materials and Methods

### 2.1. Data Collection and Variant Annotation

The reference genome was collected from the NCBI (National Center for Biotechnology Information). Other genomes were selected using the GISAID (Global Initiative on Sharing Avian Flu Data), which contains the SARS-COV-2 variants in the state of Bahia, Brazil. Genome selection took place through 1 September 2021, the period corresponding to the first and second wave of the epidemic in the state of Bahia [9]. The genomes were then aligned using the MAFFT software [10]. Finally, the first annotation and variants were analyzed using COVID-19 Genome Annotator software [11].

### 2.2. SNP Analysis

For the SNP (Single Nucleotide Polymorphisms) analysis, SNPs were classified based on mutations using Nextclade software [12]. The SNPs were classified according to impact: low impact, when synonymous mutations were added in coding regions, in initiation and stop codons; medium impact, when the mutations presented an intermediate number of private mutations and/or frameshifts; and finally, high impact, referring to those that presented a large number of mutations and/or frameshifts.

### 2.3. Genomic Sequence Analysis and Epitopes Evaluation

The genomic sequences were analyzed using the UGENE [13] and MEGA [14] software to analyze structural and functional changes, such as the number and location of transitions and transversions, and to verify the number of conserved and singleton sites present. The generated structural changes data were visualized using the ggtree v3.4.1. R package. The genomes were also verified using CorGAT software [15] to perform qualitative–quantitative analysis of the epitopes according to the genomic region of the analyzed sequences, and the Human Leukocyte Antigen (HLA) types involved.

### 2.4. Identification of Lineages and Phylogenetic Analyses

Identifications for the genome lineages (from Bahia) and the human SARS-CoV-2 reference genome from Wuhan (NC_045512.2) were obtained using Phylogenetic Assignment of Named Global Outbreak Lineages (PANGOLIN) v4.2 [6].

Phylogenetic analysis of the SARS-CoV-2 genomes was performed using the maximum likelihood (ML) method. The ML phylogenies of a large alignment (744 genomes) were inferred using IQ-Tree2 [16], with best-fitting substitution model parameters (GTR+F+R2) estimated using Model Finder with 10,000 rapid bootstrap replicates. Finally, the generated phylogenetic trees were visualized using the ggtree v3.4.1 R package [17].

## 3. Results

### 3.1. Variant Annotation

Analysis of the diversity of sequenced SARS-CoV-2 genomes from Bahia, and comparison to the reference genome (NC_045512.2), was performed for the period of 28 February

2020 to 1 September 2021. As a result, 744 genomes were obtained through a GISAID search, which included the reference genome. Furthermore, 20,773 mutations were found using the Genome Annotator, and of these it was observed that the BA-LNN03731 genome presented the highest number of mutations (47) (Figure 1).

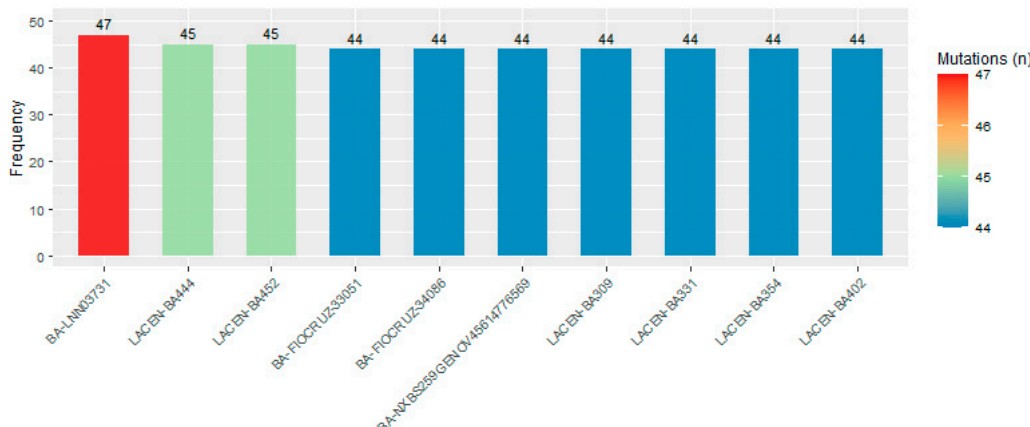

**Figure 1.** Top ten mutations in SARS-CoV-2 genomes from Bahia with the highest number of mutations through September 2021.

The nucleotide variants most present in the sample were A23403G, C241T, C3037T, and C14408T. The most frequent protein variants were S:2614G, 5′UTR:241, NSP3:F106F, and NSP12b:P314L. The majority of mutations were SNP, followed by silent and extragenic SNP, in that order (Figure 2).

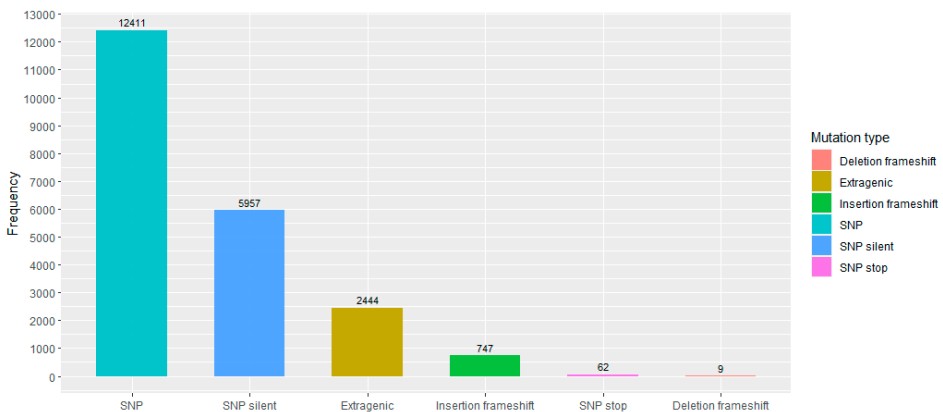

**Figure 2.** Types of SARS-CoV-2 genome mutations in Bahia through September 2021.

As for amino acids, aspartate was the most frequent (2611), followed by proline (2259) and glutamate (2128) (Figure 3). After analyzing the SNPs as to the effect of the mutations present in the genomes, the most predominant types were low-impact mutations (712), followed by medium (18), and high-impact (14) mutations (Supplementary material I).

### 3.2. Structural and Functional Analysis

During structural analysis of these genomes, 2261 variable sites were observed, with 1582 transition-type mutations: 497 synonymous, and 983 non-synonymous mutations. In addition, at 102 sites it was not possible to inform the effectiveness of the encoded amino acids. For transversion-type mutations, 679 were found, of which 78 were synonymous, 564 were non-synonymous, and 37 were indeterminate as to the encoded amino acid. In addition, 27,705 conserved sites were found, and 1383 singleton mutations.

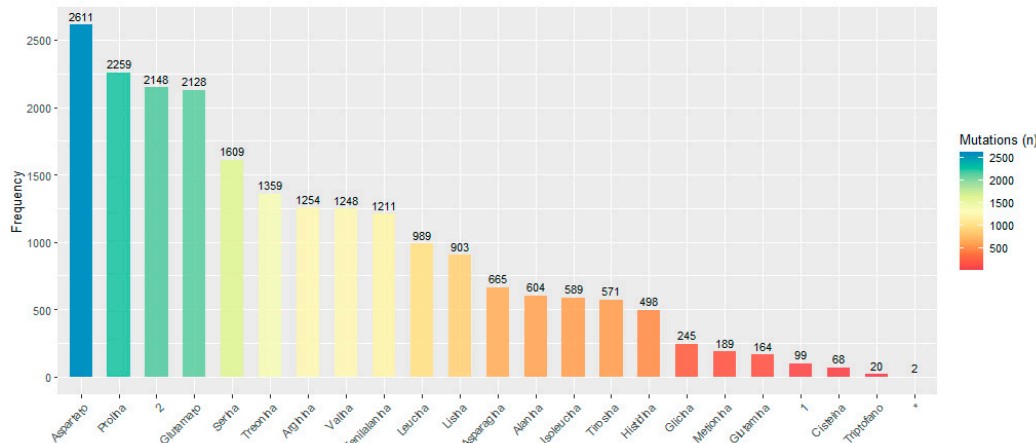

**Figure 3.** Amino acid substitution types obtained from SARS-CoV-2 genomes (originating in Bahia and presenting the most mutations) through September 2021.

Concerning the functional analysis performed using CorGAT, we obtained 9527 predictors of T cell epitopes that bind to SARS-CoV-2 proteins: 2662 for HLA-A, 3266 for HLA-B, and 3599 for HLA-C. Furthermore, epitope-associated alleles were predicted for all SARS-CoV-2 proteins except ORF3a, ORF7a, and ORF7b. For viral epitope-associated alleles, 33 were related to type I (HLA) alleles; 7 to HLA-A, 18 to HLA-B, and 8 to HLA-C types (Table 1).

**Table 1.** List of 33 HLA alleles obtained using CorGAT prediction.

| HLA Loci | Total | Alleles |
|---|---|---|
| HLA-A | 2662 | *02:01, *02:06, *11:01, *24:02, *26:01, *31:01, *33:03. |
| HLA-B | 3266 | *03:01, *04:01, *06:01, *07:02, *11:01, *15:01, *16:02, *16:06, *20:03, *22:01, *35:01, *40:02, *40:06, *44:03, *46:01, *51:01, *52:01, *54:01. |
| HLA-C | 3599 | *01:02, *03:03, *03:04, *07:02, *08:01, *12:02, *14:02, *14:03. |

### 3.3. Epitope Evaluation

The epitopes with the highest prevalence were associated with ORF1ab protein, with a total of 1769 epitopes, with HLA-A*02 (207) being the most prevalent, followed by HLA-A*11 (184). The second highest prevalence was for those epitopes associated with the Spike protein, with a total of 266 epitopes, with HLA-A*02 (32), followed by HLA-A*11 (22), and HLA-A*24 (22). In contrast, ORF6 was associated with a lower prevalence of epitopes, such as HLA-A*02 (3), HLA-A*11 (1), HLA-B*52 (2), and HLA-C*08 (1). All epitopes found presented a standard length of 9–10 amino acids (Supplementary material II).

### 3.4. Phylogenetic Analysis

Comparing the 743 genomes, PANGOLIN detected 19 groups. P.1 (361) attracted our attention, followed by P.2 (118). Of the B lineages, the highlight was B.1.1.33 (79), followed by B.1.1.28 (79). These seven groups are subdivided in the GISAID nomenclature (S, GRY, V, G, L, GH, and GR) (Figure 4) (Supplementary material III).

Using lineage classification, a phylogenetic probability tree was generated with Iqtree2 to facilitate visualization of the evolution of the genomes selected for the study. In addition to the predominance of the gamma variant, the P.2 (zeta) variant stands out; this, as with the gamma variant, originated from the B.1.1.28 lineage. Important lineages such as B.1.1.33 and B.1.1.7 (alpha variant) were also observed (Figure 5).

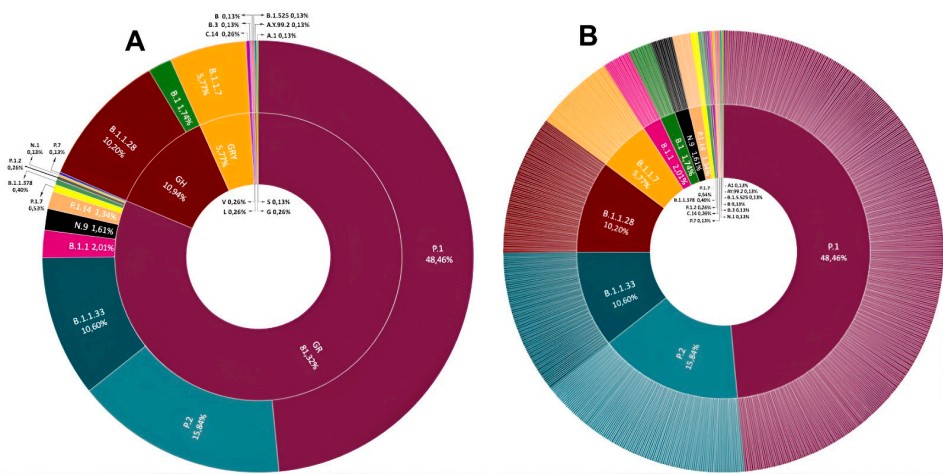

**Figure 4.** Sunburst diagrams colored in accordance with the Global Initiative on Sharing All Influenza Data (GISAID)—clades showing the relationship between GISAID and Phylogenetic Assignment of Named Global Outbreak Lineages (PANGOLIN)—annotations on the inner and outer circles, respectively for the Bahia SARS-CoV-2 genomes (*n* = 743). (**A**) The chart presents the dominant PANGOLIN proportion corresponding to each of the GISAID clades (The count for individual clades/lineages is shown in Supplementary material III). (**B**) A schematic representation of the association between the PANGOLIN lineages and the lineage's nomenclature.

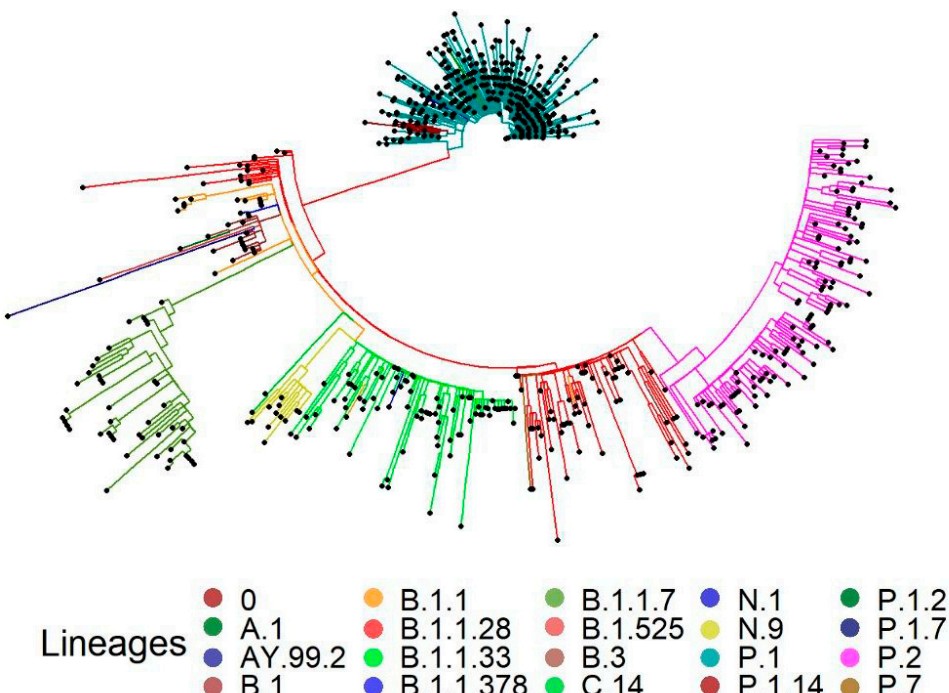

**Figure 5.** Phylogenetic tree generated using Iqtree, version 2, through the maximum likelihood model on 744 genomes (743 Bahia genomes and the reference genome), visualized using the ggtree v3.4.1 R package, and colored according to the existing lineages collected in PANGOLIN v4.2.

## 4. Discussion

Comparing the nucleotide variants found in Bahia with an Indian study [18], similar results were observed. That study analyzed 20,086 genomes and obtained the most prevalent mutations: A23403G, C3037T, C241T, and C14408T. The C14408T mutation is also implicated in a significant reduction in the rate of viral replication, and when associated

with the C241T and C3037T mutations, it further reduces virus replication. This has been analyzed using a SARS-CoV-2 replication model [19].

A divergence was found in a protein study performed in the state of Amazonas, in which the N protein variant suffered the most mutations. This structural protein forms the nucleocapsid. In the state of Bahia, the S protein (spike protein) presented the highest mutation frequency [20]. Various studies have conducted genomic analysis of SARS-CoV-2 and revealed mutations in many genes, including ORF1ab, ORF3a, ORF6, ORF7, ORF8, ORF10, S, M, E, and N. Among them, the ORF1ab, S, and ORF8 genes were reported as having considerably more mutations than other genes [5].

The mutations found in amino acids allow for explaining essential mechanisms of action of the virus concerning the host. For example, the high prevalence of aspartate and glutamate substitutions explains the increased affinity of the virus (when aspartate is replaced by tyrosine) for ACE-2 cell receptors. The rapid epidemic expansion of SARS-CoV-2 can thus be explained [4].

In the structural and functional distribution of these genomes, the variable sites were transition and transversion types, with non-synonymous mutations prevailing over synonymous in both. This was also observed in a global study that analyzed 4254 strains of SARS-CoV-2, to obtain 767 synonymous mutations against 1352 non-synonymous mutations. However, they did not specify whether the mutations were of the transition or transversion type [20].

When analyzing high-impact mutations, there was a lack of information in the literature concerning other possible mutation locations (or resulting viral advantages) present in the Brazil/BA-FIOCRUZ-PVM54698/2021|EPI_ISL_2663286|2021-03-04 genome. The set of mutations is believed to be exclusive to Bahia. The Brazil/BA-FIOCRUZ-PVM54698/2021 |EPI_ISL_2663286|2021-03-04 genome presents three (3) labeled private mutations and 20 unlabeled private mutations. Labeled private mutations are private mutations in a common genotype within a clade, while unlabeled private mutations are private mutations that are neither labeled nor reverse [21].

Other high-impact mutations were found in various countries such as Japan, the United States, and France [22–25]. The advantages of these mutations are reported to be decreased susceptibility to treatments and monoclonal effects of bamlanivimab/etesevimab, decreased neutralization for convalescent and post-vaccination sera, and increased production of Spike proteins (ACE2), which increases both transmissibility and virulence [26,27].

HLA is essential in both antigen presentation and adaptive immune response [28]. Identifying alleles associated with viral epitopes is therefore essential for developing vaccines and therapeutic options, as well as for developing a better understanding of the mechanism of viral infection. For example, HLA-B*15:01 (also observed in other studies) is strongly associated with asymptomatic SARS-CoV-2 infection, and is likely involved in early viral clearance [29]. HLA-A*02:01 is protective against severe cases of COVID-19 [30].

During lineage analysis, the prevalence of the P.1 and P.2 lineages was observed in more than 50% of the genomes, this was also seen in the greater Brazilian scenario, where these same lineages (after October 2020) reached 75% in national-level sequencing [4].

According to data from the FIOCRUZ Genomic Network, through October 2020, the B.1.1.28, and B.1.1.33 strains were the most prevalent in Brazil, and played an essential role in the first wave of the pandemic [4,31]. This was corroborated in the Bahia scenario, since these lineages were initially prevalent, and both originated from the B.1.1 lineage. From the phylogenetic tree, we find the lineage B.1.1.7, alpha variant arising from the United Kingdom, its outstanding mutation is the amino acid exchange of asparagine with tyrosine [32].

As seen in this phylogeny, the gamma variant is predominant. Both the gamma and zeta variants originate from the B.1.1.28 lineage, with the gamma variant appearing in Manaus and the zeta variant appearing in Rio de Janeiro. Both variants present protein S mutations and certain SNP types, in which glutamate is converted to lysine [4]. These mutations, in particular, affect both transmissibility and host immune response [33], and are

evidently one of the reasons for the high frequency of the P.1 lineage in this population. Yet, when compared with greater Brazil, in March 2021, Bahia was the state with the seventh highest preponderance of gamma lineage genomes [32].

According to the WHO, these variants can be divided into variants of concern (VOCs) and variants of interest (VOIs). VOCs present increased transmissibility and virulence, while VOIs involve community transmission or are present in multiple countries. Of the existing VOCs in the period of the present study, only the alpha and gamma variants were present in the state of Bahia. When comparing existing VOIs from March 2020 to September 2021 (in addition to the zeta variant, which stood out in frequency), the presence of the beta variant (B.1.525) was noted [34].

## 5. Conclusions

Our study was restricted to the period from March 2020 to September 2021. This time window denied viewing strains that might have emerged afterward, and consequently, new mutations. However, the results and analysis presented herein permit an enriched epidemiology for the circulating genomes of this virus in Bahia and Brazil, in addition to providing information and bases for developing new therapies against COVID-19.

**Supplementary Materials:** The following supporting information can be downloaded at: https://www.mdpi.com/article/10.3390/covid3050047/s1, Supplementary material I: Private mutations identified in 743 samples using Nextclade. The number of mutations found at each position, the locale, and biological advantages are shown; Supplementary material II: Representation of epitope sequences and their HLA alleles in the analyzed SARS-CoV-2 genomes; Supplementary material III: Distribution of the Bahia sequences (*n* = 743) from the Global Initiative on Sharing All Influenza Data (GISAID), and Phylogenetic Assignment of Named Global Outbreak Lineages—clades (PANGOLIN).

**Author Contributions:** Conceptualization, I.C.; methodology, G.A., G.M. and L.C. software, G.A. and A.S.; validation, G.A., G.M. and I.C.; formal analysis, G.A.; investigation, G.A. and G.M.; data curation, I.C.; writing—original draft preparation, G.A., G.M., L.C., J.d.C.-N. and J.S.; writing—review and editing, I.C.; visualization, A.S.; supervision, A.S.; project administration, I.C.; resources, A.S. and I.C. All authors have read and agreed to the published version of the manuscript.

**Funding:** This research received no external funding.

**Institutional Review Board Statement:** Not applicable.

**Informed Consent Statement:** Not applicable.

**Data Availability Statement:** The data used to support this study are included in this paper.

**Conflicts of Interest:** The authors declare no conflict of interest.

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
