# Peer review of "Genetic Analysis and Epitope Prediction of SARS-CoV-2 Genome in Bahia, Brazil: An In Silico Analysis of First and Second Wave Genomics Diversity"

_covid, doi:10.3390/covid3050047_

Round 1

Reviewer 1 Report

Andrade et al., reports the genome analysis of circulating SARS-2 variants in the Brazilian state of Bahia. While I consider the study to be important, there are some missing clarifications and some limitations that I could find. Following are my comments.  

Why did the authors exclude prediction of epitope-associated alleles for ORF3a, ORF7a and ORF7b? Please clarify.

What is the meaning of "private mutations"? Please clarify.

The authors need to elaborate on each of the 14 high impact SNPs detected in terms of where do these mutations map to. Which proteins are affected. If some of the mutations fall in the intergenic region, how such mutations provide a survival advantage wrt to the non-mutated counterpart. Are any of these 14 SNPs detected in the currently circulating variants worldwide or are they exclusive to Bahia?

Another important aspect is whether certain high impact SNPs are frequently encountered in certain geographical regions of the world. Although, this would involve a lot more in-depth analysis, I would encourage the authors to mention few lines in the discussion.

Author Response

Dear Dr,

Kind regards,

Reviewer 2 Report

In the manuscript “Genetic analysis and epitopes prediction of SARS-CoV-2 genomes from the state of Bahia, Brazil: an in-silico approach” the authors focus on the recent SARS-CoV-2 pandemic. They are investigating the genomic diversity of SARS-CoV-2 strains circulating in Bahia. They selected 744 genomes of which approximately 20,000 mutations were identified, mainly SNP. The authors demonstrated that the SARS-CoV-2 variants isolated in Bahia clustered closely. These findings could contribute to increased vaccine effectiveness.

The manuscript is very interesting and well-organized. However, I suggest reading the paper written by Zannella et al. (Design of Three Residues Peptides against SARS-CoV-2 Infection; PMID: 36298659).

Author Response

Dear Dr,

Kind regards,

Reviewer 3 Report

The authors just selected data upto Sep, 2021. This makes it too outdated. Please consider adding uptil latest.

Put more focus on variants.

Author Response

Dear Dr,

Kind regards,

Round 2

Reviewer 1 Report

The authors have addressed all the concerns. 

Author Response

Dear reviewer,

The responses are attached.  

Kind regards,

Reviewer 2 Report

the authors improved in part  the quality of the manuscript which is now suitable for publication.

Reviewer 3 Report

Authors need to mention in title that the study corresponds to first and second wave transmission strains.

Language is incomprehensible at places an dtext doesnot make good sense e.g. 'Therefore, it is noticeable that the pandemic has brought challenges that need to be 54 faced, and bioinformatics plays a key role in this process. The approach in this area is 55 found in obtaining and analyzing important parameters, such as the virus transmissibility 56 rate, in the reconstruction of transmission routes and the identification of possible animal 57 sources and reservoirs [7], ' needs editing

break methods into sections and add more headings. Same for results

Conclusion should be under separate heading.

Author Response

(The authors gave the same response as above.)

Round 3

Reviewer 3 Report

Good job. All my points have been addressed. I recommend acceptance.